# Dissecting Temporal Understanding in Text-to-Audio Retrieval

## ABSTRACT

Recent advancements in machine learning have fueled research on multimodal interactions, such as for instance text-to-video and text-to-audio retrieval tasks. These tasks require models to understand the semantic content of input videos, including objects, sounds and characters. The models also need to learn their spatial arrangement and the temporal relationships of sounds. In this work, we tackle the temporal ordering of sounds, which is an understudied problem in the context of text-to-audio retrieval. In particular, we dissect the temporal understanding capabilities of a state-of-the-art model for text-to-audio retrieval on the AudioCaps dataset. Additionally, we introduce a synthetic text-audio dataset that provides a controlled setting for evaluating the temporal understanding of recent models. Lastly, we investigate a new loss function that encourages text-audio models to focus on the temporal ordering of events.

## CCS CONCEPTS

• **Information systems** → **Speech / audio search**; **Multimedia and multimodal retrieval**.

## KEYWORDS

text-to-audio retrieval, temporal understanding

**ACM Reference Format:**
Anonymous Authors. 2024. Dissecting Temporal Understanding in Text-to-Audio Retrieval. In *Proceedings of the 32nd ACM International Conference on Multimedia (MM'24), October 28-November 1, 2024, Melbourne, Australia.* ACM, New York, NY, USA, 7 pages. https://doi.org/10.1145/nnnnnnn.nnnnnnn

## 1 INTRODUCTION

The continued improvement of model capacities and the increase in data available have led to impressive results on multimodal tasks, such as text-image understanding [23] and text-audio understanding [8]. In the domain of text-audio understanding, tasks include text-to-audio retrieval [3, 13, 17, 19, 26, 30, 38], audio captioning [6, 7, 15] and recently, text-to-audio generation [11, 14, 37]. Understanding details, such as temporal ordering of events, is important if we want our systems to give the best search results or generate reliable content for a text query. Recently, [29] showed that text-audio models do not use temporal cues available in text-audio datasets.

In this work, we build on [29] and examine limitations of current state-of-the-art text-audio models, particularly in their utilization

of temporal information. Different from [29] who use a Convolutional Neural Networks (CNNs) based audio encoder, our analysis uses the recent transformer-based audio encoder HTS-AT [2] that serves as a component of state-of-the-art text-to-audio retrieval models [17, 30]. We assess whether using an attention mechanism-based audio encoder, instead of a CNN-based one, improves the temporal understanding of text-audio models. Additionally, we investigate in detail the experimental designs and datasets used to determine if poor temporal understanding in current state-of-the-art models is caused by the training data or by the model architecture.

To determine whether commonly used text-audio datasets, such as AudioCaps [12], are suitable for training and evaluating current models' ability to comprehend time, we examine the relative distribution of audio descriptions that contain temporal cues, plotting their frequency in relation to the total number of descriptions. Our analysis shows that the AudioCaps dataset suffers from biases caused by the way humans describe events. That is, we tend to describe events in the order they appear. When first hearing the sounds of a dog barking and then the sound of a human speaking, we describe this as 'A dog barking followed by a human speaking' rather than 'A dog barking before a human speaks'. To try to address the lack of *some* temporal examples, in [29], the authors generate new text-audio pairs starting from their existent text-audio data. They concatenate the audio files in a specific order, and then generate a description that reflects that e.g. if the generated sound is $Sound_1 + Sound_2$, the description is '<Original description of $Sound_1$> before <Original description of $Sound_1$>'. They thus further increase the training size of the data by 40%. In comparison to them, we rephrase existing text descriptions such that to preserve the content but use a more uniform distribution of textual temporal cues. We investigate the impact of employing a more uniform set of training examples on the performance outcomes of models, comparing text-to-audio retrieval results on the original test data with those on rephrased test data.

Furthermore, we present an empirical evaluation of the correctness and completeness of AudioCaps descriptions by leveraging Large Language Models (LLMs) as oracles[1]. More specifically, we provide an LLM with the original descriptions and with the subset of AudioCaps for which we have temporally localised sounds (provided by [33]). We ask the LLM to classify the sentences into correct – if the description contains all the sounds and the correct ordering, incomplete – if the description is missing sounds or is missing temporal context, and incorrect – if the description contradicts the provided grounded sounds. We observe that about 23% of the descriptions are incomplete or incorrect. This can contribute to models trained on AudioCaps not understanding temporal ordering.

To gain further insights into the temporal understanding capabilities, we propose a synthetic dataset that provides a controlled setting for analysing text-audio models. This dataset contains only 10 second long audios, keeping in line with the general setting that

---

[1]An oracle in a computational context is a theoretical construct that provides perfect answers or decisions.

current models have been trained on. We show that the considered model struggles to use temporal cues in the synthetic dataset, too, confirming the findings from [29] in a controlled setting. This is useful as it allows us to decouple the bad temporal performance of the model from the data not being suited for the task. Lastly, we propose a simple text-based contrastive loss function and show that it results in the model paying more attention to the temporal ordering of events. This also gives improvements in the overall retrieval results on the synthetic dataset.

In summary, we make the following contributions: (i) We show why existent text-to-audio retrieval datasets are not good indicators of a text-audio model's ability to understand temporal ordering, (ii) We propose a more uniform version of AudioCaps that is better suited for the temporal understanding task. We provide benchmarks and an analysis of the behaviour of current models on the original and more uniform versions of this dataset. This version keeps the audios intact and only requires changing the text descriptions. (iii) We propose a synthetic dataset and use it to evaluate the model's understanding of time, (iv) We investigate an additional loss term to encourage the model to focus on text-based temporal cues.

## 2 RELATED WORK

**Text-to-audio retrieval.** Text-to-audio retrieval involves matching a textual query with its most relevant audio file. This task of searching through audio databases can be approached in multiple ways. One simple way is to match the text query with the title or the metadata of the audio file, provided it exists. However, for unlabelled databases, the aim is to find an audio file that has the content specified by the user through the given text query. This is called semantic search. For many years, text-audio semantic retrieval has used audio class labels made of individual or few words as text queries [9, 10, 25, 28]. More recently, [13, 19] proposed new benchmarks where the text query is a free-form text description rather than a pre-defined class label, allowing for more control over the retrieved audio content. Collecting new text-audio pairs for training and using state-of-the-art transformer-based audio encoders has proven beneficial on the text-to-audio retrieval benchmarks [17, 30, 38]. As the annotation of audio files with descriptions is time consuming, some of the text-audio pairs collected by [30] and [17] contain short audio labels instead of descriptions. To overcome this, [30] employed the T5 [24] model to generate proper descriptions starting from audio labels, whilst [17] used ChatGPT [20]. [17] also used ChatGPT to clean audio descriptions from datasets such as BBC Sound Effects[2] by removing visual-based content. Another line of works considered metric learning objectives for text-to-audio retrieval [16, 32]. Other concurrent research pushed the text-to-audio retrieval results even further by training models with additional modalities, such as video and speech [4, 27]. Recently, [18] introduced new text-to-audio retrieval benchmarks on egocentric video data.

**Text-audio grounding.** [33] proposes a new set of data annotations for a subset of the AudioCaps dataset [12], with the aim of grounding each sound to a time interval. For this, annotators labelled the start and end times of all relevant sounds in each audio

---

[2]https://sound-effects.bbcrewind.co.uk/

clip. [34, 35] investigated the task of weakly supervised text-to-audio grounding. The audio grounded dataset has also been used for learning to align sounds and text in an unsupervised manner [31]. [1] used the grounded sounds to propose new metrics for audio captioning. In this work, we use this subset to provide an empirical evaluation of the quality of existing AudioCaps captions. More specifically, we provide these grounded sounds and the corresponding AudioCaps descriptions to an LLM and ask it to evaluate if the descriptions are correct and complete.

**Text-to-audio retrieval with synthetic data.** A concurrent work [36] claims that commonly used text-audio datasets only contain simple audio descriptions that are not always complete. In particular, they lack temporal cues, the number of times a sound can be heard, or details about sounds overlapping. [36] propose a synthetic dataset for audio captioning, by merging 'atomic' sounds in a controlled way. We also generate a synthetic dataset and use it to analyse the temporal understanding capabilities of text-audio models.

**Temporal understanding of text-audio models.** [29] show that text-audio models do not pay attention to temporal cues in text queries, such as 'followed by', or 'after'. One example of an experiment employed by [29] for checking if models understand time, is replacing temporal cues with words that represent a wrong ordering, e.g. replacing 'then' with 'as'. Then, the model's performance on the 'wrong' descriptions is evaluated, finding that this performance is similar to when the temporal ordering in the text queries is correct. In their study, [29] utilize Convolutional Neural Networks (CNNs) for audio processing. They identify a critical limitation of CNN-based models: the practice of applying temporal pooling across all embeddings can result in the loss of temporal information. To try mitigating this issue, they suggest augmenting the CNN architecture with several transformer layers to preserve temporal dynamics. In contrast, contemporary models built on transformers, inherently incorporate mechanisms to handle temporal data more effectively. Different to [29], we investigate the temporal understanding of a current transformer-based state-of-the-art audio-text retrieval model. In particular, we analyse if a transformer-based model also ignores temporal cues. We dive deep into the analysis of descriptions in text-audio datasets in the context of temporal understanding. Additionally, we propose a synthetic text-to-audio retrieval dataset and perform temporal understanding experiments on it. Lastly, the approach proposed by [29] for helping models better understand time does not improve the overall performance on downstream retrieval benchmarks. In this work, we investigate a different way to guide the model to focus on temporal cues.

## 3 TEMPORAL UNDERSTANDING IN TEXT-TO-AUDIO RETRIEVAL

### 3.1 AudioCaps dataset

The AudioCaps [12] dataset contains paired audio clips and text descriptions. The training set consists of one text description for each audio file. The validation and test sets contain five descriptions for each audio file. In this setting, which is employed by all benchmarks utilising AudioCaps, if any of the five text descriptions matches with the audio clip, this corresponds to 100% retrieval accuracy.

In this section, we take a closer look at the AudioCaps dataset, which is employed in all related text-to-audio retrieval works for

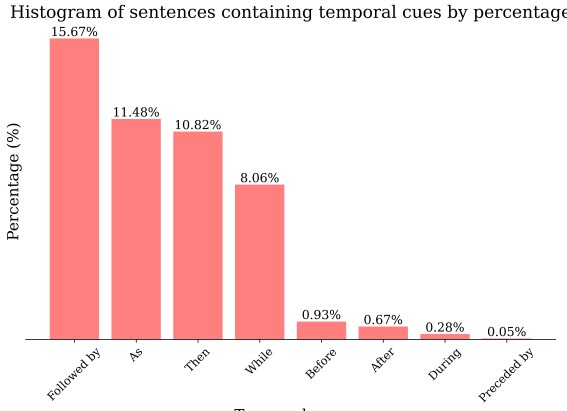

Figure 1: Distribution of temporal conjunctions and prepositions in the full AudioCaps dataset.

training and evaluation. We want to better understand the temporal characteristics of this dataset to gauge if the data available for training text-to-audio retrieval models is a part of the problem of models not understanding temporal cues [29].

First, we analyse the distribution of temporal conjunctions and prepositions in the audio captions in the AudioCaps dataset in Fig. 1. We observe that most conjunctions and temporal prepositions suggest future events, i.e. 'Followed by', 'Then'. This is closely followed by the joint occurrence of audio events, i.e. 'As' and 'While'. However, almost no examples contain the temporal prepositions 'Before' or 'Preceded by' which is reasonable as humans would not naturally describe events in that order. A similar analysis is performed by [29], where they provide percentage distribution for 'Followed by', 'Then', 'Before' and 'After'. In [29] this distribution describes their training and test data, which are formed form a combination of multiple datasets amongst which AudioCaps [12] and Clotho [7]. Here, we consider all words in the AudioCaps descriptions that represent temporal ordering. Given the distribution in Fig. 1, expecting a model trained on this data to understand the meaning of reverse temporal prepositions is unreasonable. At the same time, the test data also suffers from the same problem, therefore, using AudioCaps benchmarks for deciding if models understand temporal ordering is not optimal either.

Next, we *empirically* evaluate the correctness and completeness of AudioCaps descriptions by using the grounded sound time intervals provided by [33]. Through manual inspection, we notice that many AudioCaps descriptions are composed of multiple sounds. For instance, a 10 second audio file with a bird singing from second 0 to second 6 and a dog barking from second 4 until second 10 can be described as 'Bird singing and/as dog barks'. Alternatively, this could be described as 'Bird singing followed by dog barking'. Both descriptions are correct, however, a more complete version of these descriptions would be, for example, 'Bird singing, soon joined by a dog barking. Their sounds overlap briefly before the bird stops, while the dog continues barking.'. If the description is not complete, however, how could a model learn the difference between 'as' and 'followed by' when they describe the same audio clip?

To *empirically* evaluate the completeness and quality of the descriptions in AudioCaps with grounded sound sources, we use an LLM, specifically GPT-4 [21]. We provide the LLM with the AudioCaps description, the grounded sources and their time intervals. We use one-shot prompting to give the model an example, such that it better understands the task. We then ask the model to provide an evaluation of 'correct', 'incomplete' and 'wrong' for the AudioCaps description based on the sound sources information. Details for our prompt are shown in Tab. 1. We process the outputs of the LLM, yielding proportions of correct, incomplete, and wrong descriptions in the subset of AudioCaps presented in Tab. 2. On average, 23% of the descriptions are incomplete or wrong. This percentage increases for descriptions containing *future* and *past* temporal cues. The use of *future* and *past* refers to the fact that if 'Sound 1' and 'Sound 2' are connected by a *future* temporal cue, then that means that 'Sound 1' comes first and is followed by 'Sound 2'. If a *past* cue is used, then 'Sound 1' comes after 'Sound 2', e.g. 'Bird sings after dog barks'. *Future* cues include 'Followed by', 'Before' and 'Then', e.g. 'Bird sings before dog barks'. For *past*, we consider 'Preceded by' and 'After'. Based on the significant proportion of incomplete or wrong descriptions, and the distribution of temporal textual cues, we conclude that AudioCaps is not well-suited for analysing if text-audio models understand temporal ordering.

## 3.2 Model performance on AudioCaps

In this section, we investigate the performance of a state-of-the-art model for text-to-audio retrieval on AudioCaps in detail.

*3.2.1 Evaluation metrics.* Throughout all experiments, we use the standard evaluation metrics for retrieval: recall at rank $k$ (R@$k$). This measures the percentage of targets retrieved within the top $k$ ranked results. Higher numbers are better. We report results for text-to-audio (T → A) and audio-to-text retrieval (A → T). We report the mean of three runs that use different random seeds.

*3.2.2 Model.* We employ the state-of-the-art text-audio model by [17], utilising an HTS-AT audio encoder [2], and a pre-trained BERT encoder for text. After encoding audio and text, an MLP projects the embeddings into the same space. We use the model variant pre-trained on WavCaps [17]. In our experiments, we fine-tune the model for 40 epochs and use the same setup as [17]. The best model is selected based on the highest average validation retrieval accuracy R@1.

*3.2.3 Loss function.* We use the same loss as [17] - a normalised temperature scaled bidirectional cross-entropy loss (NT-Xent) [5]. We call this loss $\mathcal{L}_{at}$ with

$$s_{ij} = \frac{f(a_i) \cdot g(t_j)}{\|f(a_i)\|_2 \|g(t_j)\|_2}, \tag{1}$$

$$\mathcal{L}_{at} = -\frac{1}{2B} \sum_{i=1}^{B} \left[ \log\left( \frac{\exp(s_{ii}/\tau)}{\sum_{j=1}^{B} \exp(s_{ij}/\tau)} \right) + \log\left( \frac{\exp(s_{ii}/\tau)}{\sum_{j=1}^{B} \exp(s_{ji}/\tau)} \right) \right]. \tag{2}$$

Here $f(\cdot)$ is the audio encoder and $g(\cdot)$ the text encoder. $s_{ij}$ is the cosine similarity, $a_i$ is an audio input, $t_j$ is a text input, $B$ is the batch size, and $\tau$ is a temperature parameter. More details in [17].

*3.2.4 Data used.* For our analysis, we construct a more *uniform* version of the AudioCaps dataset with descriptions having a more balanced distribution of temporal conjunctions and prepositions.

**Table 1: Methodology for evaluating the quality of the grounded subset of AudioCaps using an LLM. Our input prompt includes setting the scene, one-shot prompting with an example, followed by the generation of new examples.**

| Prompt |
|---|
| Given descriptions of audio files and detailed temporal information about specific sounds within these files, where a sound may be present during multiple, distinct time intervals, your task is to evaluate the accuracy of each description with a primary focus on the timing and sequence of these sounds. Each audio file is 10 seconds long. For every description, assess its accuracy specifically in terms of how well it captures the chronological order and exact timing of sounds. Classify your evaluation into one of three categories: 'Correct', 'Incomplete', or 'Wrong'. If necessary, provide a corrected description that not only fixes inaccuracies related to timing but also maintains the original writing style of the description. Your analysis should critically examine the temporal details provided, ensuring your assessment is primarily guided by the accuracy of these temporal sequences. |
| Keep in mind the following: Pay attention to whether the description matches the start and end times of sounds accurately. Consider if the sequence of described sounds follows the actual sequence in the audio file. Evaluate if the description misses any sounds within the specified time frames or includes sounds that do not occur within these times. Use similar vocabulary as the original audio description. |
| Example: Input: Original audio description: A power tool motor running then revving Localized components and their start and end times: revving: 2.154, 10.02; a power tool motor running: 0.0, 10.02; Output: Evaluation: Incomplete Corrected description: A power tool motor running throughout, with revving starting early on and continuing alongside the motor's running sound until the end. |

**Table 2: Proportion of correct, incomplete and wrongly captioned AudioCaps data as determined by an LLM. First row contains the total numbers of grounded descriptions. The other rows show proportions for specific temporal cues.**

| Preposition | Correct | Incomplete | Wrong |
|---|---|---|---|
| Total (#) | 3835 | 636 | 503 |
| As (%) | 75.3 | 13.9 | 10.8 |
| Followed by (%) | 60.6 | 15.3 | 24.1 |
| Then (%) | 62.1 | 15.9 | 22.0 |
| While (%) | 72.0 | 15.4 | 12.6 |
| Before (%) | 58.8 | 9.8 | 31.4 |
| After (%) | 54.5 | 6.1 | 39.4 |
| Proceeded by (%) | 50.0 | 0.0 | 50.0 |
| During (%) | 53.3 | 13.3 | 33.3 |
| And (%) | 75.6 | 13.5 | 10.9 |

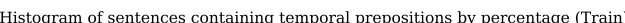

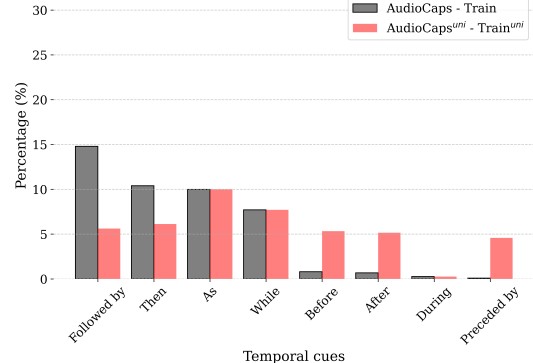

**Figure 2: Distribution of temporal conjunctions and prepositions in AudioCaps training data. We compare the proportion of temporal textual cues in the original training dataset (Train) and the more uniform dataset ($Train^{uni}$).**

In particular, we rephrase AudioCaps descriptions to preserve the original meaning while varying the use of temporal conjunctions and prepositions. We investigate if this helps with temporal understanding. Specifically, we re-write the descriptions from the original AudioCaps dataset to generate the $AudioCaps^{uni}$ dataset with corresponding $Train^{uni}$, $Val^{uni}$ and $Test^{uni}$ subsets. There are two approaches to re-writing the descriptions. One is to replace the temporal cues with something that has the same meaning e.g.

'Bird singing followed by dog barking' is equivalent to 'Bird singing before dog barking'. The second approach is to re-order the text location of events and also change the temporal cue e.g. 'Bird singing followed by dog barking' becomes 'Dog barking after bird singing'. We present the distribution of temporal cues in the original AudioCaps dataset and its uniform variant in Fig. 2 and Fig. 3. An analysis of the validation split can be found in the supplementary material.

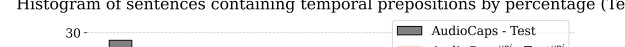

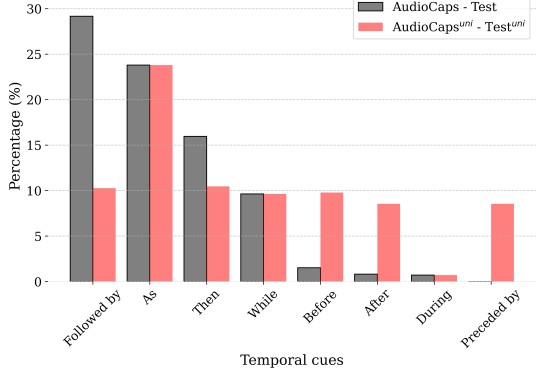

**Figure 3: Distribution of temporal conjunctions and prepositions in the AudioCaps test dataset.**

In addition to the more uniform AudioCaps version, we create a test subset where at least one of the 5 text descriptions contains *future* and *past* temporal cues (as defined in Sec. 3.1). We do this to evaluate the performance of the model on sentences that actually contain temporal cues of interest. We call this subset *TempTest*.

*3.2.5 Experiments.* We consider three main experiments. First, we conduct the standard evaluation for text-to-audio retrieval on the AudioCaps test set. The performance on the standard test set serves as a point of reference for the training and evaluation on different variants of the AudioCaps data.

The second experiment involves reversing the ordering of sounds in the text queries of the test set. We call this $Test^{rev}$. The purpose of this experiment is to see what happens if the temporal text descriptions keep the same temporal preposition or conjunction but the sound sources are reversed, resulting in a wrongly ordered description, e.g. *Birds singing before dog barks* becomes Dog barks before birds singing. If the model understands the temporal ordering of events, the performance of the model should drop for the 'wrongly' ordered events as compared to the original test set performance.

For the third experiment, we replace the temporal cue in a description with its opposite, thus changing the order of events without changing their position, e.g. *Birds singing before dog barks* becomes *Birds singing after dog barks*. We refer to this as $Test^{rep}$. More concretely, we do the following replacements: ('followed by'→'preceded by'), ('preceded by'→'followed by'), ('after'→'before'), ('before'→'after'), and ('then'→'before').

If the model does not understand temporal cues, we expect it to perform similarly well on *Test*, $Test^{rev}$ and $Test^{rep}$. Conversely, if it understands temporal ordering, *Test* performance should be considerably higher than $Test^{rev}$ and $Test^{rep}$. We would also expect $Test^{rev}$ and $Test^{rep}$ to be similar, as the meaning of the sentence is the same but opposite of the *Test* sentences meaning. We additionally consider *rev* and *rep* subsets of the temporal subset *TempTest*.

In Tab. 3, we take the checkpoint provided by [17] which was trained on WavCaps [17] and finetune it on the original AudioCaps training dataset. We notice that on the reversed $TempTest^{rev}$ set the model performs worse, indicating that the model understands

**Table 3: Text-to-audio retrieval and audio-to-text-retrieval results on the AudioCaps and AudioCaps$^{uni}$ datasets for the model fine-tuned on AudioCaps (Train). We report retrieval accuracies R@1. Reverting the order of events (generally) does not change performance.**

| Eval Dataset | Subset | T→A | A→T |
|---|---|---|---|
| | | R@1 | R@1 |
| AudioCaps | Test | 43.71 | 56.57 |
| | TempTest | 50.51 | 63.74 |
| | TempTest$^{rev}$ | 43.90 | 57.71 |
| | TempTest$^{rep}$ | 49.55 | 62.67 |
| AudioCaps$^{uni}$ | Test | 41.54 | 53.84 |
| | TempTest | 48.61 | 61.37 |
| | TempTest$^{rev}$ | 47.37 | 62.10 |
| | TempTest$^{rep}$ | 47.60 | 60.89 |

**Table 4: Text-to-audio retrieval and audio-to-text retrieval results on the AudioCaps$^{uni}$ dataset for the model fine-tuned on AudioCaps$^{uni}$ (Train$^{uni}$). Improved results on Test$^{uni}$. Slightly bigger drop in *rev* and *rep* wrt TempTest.**

| Eval Dataset | Subset | Loss | T→A | A→T |
|---|---|---|---|---|
| | | | R@1 | R@1 |
| AudioCaps$^{uni}$ | Test | $\mathcal{L}_{ta}$ | 43.67 | 53.88 |
| | TempTest | $\mathcal{L}_{ta}$ | 50.67 | 61.31 |
| | TempTest$^{rev}$ | $\mathcal{L}_{ta}$ | 46.82 | 59.31 |
| | TempTest$^{rep}$ | $\mathcal{L}_{ta}$ | 47.45 | 59.06 |

temporal ordering. However, on *TempTest$^{rep}$* which contains the replacement of temporal cues, the model performs similarly to on *TempTest*. This is interesting, as the temporal ordering of both $TempTest^{rev}$ and $TempTest^{rep}$ is reversed and wrong as compared to *TempTest*. The only difference is that for the former, the actual positional text locations of the sounds are swapped, whilst for the latter the meaning is reversed by changing the temporal connector. This leads us to believe that at best, the model learns text location-based ordering rather than the ordering given by the text connector.

We then run the same experiments on the test sets of AudioCaps$^{uni}$. We notice that the model is unable to identify the text-based order of sound events, with results on all 'correct' (TempTest) and 'wrong' (TempTest$^{rev}$ and TempTest$^{rep}$) splits being almost the same.

Next, we investigate if the model does not understand temporal ordering due to a lack of variety in the training examples. We take the same pre-trained checkpoint as before [17], and finetune it on the *AudioCaps$^{uni}$ Train$^{uni}$* set. We notice that the overall performance on the *AudioCaps$^{uni}$* test sets and the corresponding temporal subsets is higher when finetuning on a more uniform distribution of temporal cues (Tab. 4) than when finetuning on the original training data (Tab. 3). Thus, the lack of understanding temporal ordering is in part due to the training data not containing examples of *past* temporal cues. We also notice some signs of better temporal understanding, with a slightly bigger drop in performance on the *TempTest$^{rev}$* and *TempTest$^{rep}$* sets relative to *TempTest*.

**Table 5: Text-to-audio retrieval and audio-to-text retrieval results on the SynCaps dataset for the model fine-tuned on SynCaps with the $\mathcal{L}_{ta}$ (audio-text alignment) loss function.**

| Subset | T→A | A→T |
|---|---|---|
| | R@1 | R@1 |
| Test | 67.70 | 65.50 |
| Test$^{rev}$ | 66.67 | 64.95 |
| Test$^{rep}$ | 67.08 | 63.64 |

## 4 TEMPORAL UNDERSTANDING IN CONTROLLED SETTING

We analyse the text-audio model's temporal understanding in a controlled setting where we can guarantee correct alignment of text-audio pairs.

### 4.1 Data generation

We use the ESC-50 [22] environmental sound classification dataset to generate a synthetic dataset for text-to-audio retrieval with a focus on temporal understanding capabilities. ESC-50 is a dataset of 2000 audio samples from 50 classes. As this dataset is clean and contains clear 'atomic' sounds (i.e. 5 second audios containing only one sound), we use it for synthetic data generation.

We first ask an LLM to take the sound labels from ESC-50 and generate textual descriptions in the style of AudioCaps (e.g. 'dog'→ 'dog barking'). To generate the text-audio pairs, we take two sounds and their LLM-generated labels and concatenate them based on the temporal order we decide on. We call this dataset *SynCaps*.

To avoid any confusion, we only use *future* and *past* temporal cues. This is because synchronous temporal cues such as 'as' or 'during' can represent many things, especially in a noisily labelled dataset. They can be used for sounds that completely overlap, or for partial overlaps of sounds, ignoring the actual order in which the sounds appear. The test set contains unique sound components that are not used in the training and validation sets. This leads to 485 examples of 10 second long audio clips. For training and validation, we allow the same 5 seconds sound component to appear on average 5 times and apply 5 types of augmentation, to reduce overfitting on the training and validation sets. We use augmentations, such as time shifting, volume adjustment, pitch shift, time stretch, and added noise. We also allow an overlap between the files of up to 1 second. This results in a total of 4400 training files and 485 validation files.

### 4.2 SynCaps Experiments

We analyse the temporal understanding of the text-audio model in the more controlled setting of the SynCaps dataset. For this, we take the same pre-trained model from [17] and finetune it on SynCaps.

We see that evaluating on the 'reversed'(*rev*) and 'replaced'(*rep*) datasets gives almost the same results as using the correct (original) test data (Tab. 5). This shows that the model indeed does not understand temporal cues even on a simple dataset.

**Table 6: Text-to-audio retrieval and audio-to-text retrieval results on the SynCaps dataset for the model fine-tuned on SynCaps using our text-text contrastive loss $\mathcal{L}_{tt}$.**

| Subset | Loss | T→A | A→T |
|---|---|---|---|
| | | R@1 | R@1 |
| Test | $\mathcal{L}_{ta} + \lambda\mathcal{L}_{tt}$ | **69.83** | **71.13** |
| Test$^{rev}$ | $\mathcal{L}_{ta} + \lambda\mathcal{L}_{tt}$ | 40.41 | 43.43 |
| Test$^{rep}$ | $\mathcal{L}_{ta} + \lambda\mathcal{L}_{tt}$ | 44.95 | 47.70 |

### 4.3 Proposing new loss

We propose a loss function $\mathcal{L}_{tt}$ that enhances the understanding of temporal information. It is formulated as a text-to-text contrastive loss, which relies on pairs of positive examples (have the same temporal significance as the original sentence) and negative text examples (have the opposite temporal meaning). Concretely, given the original description *Bird sings followed by dog barks*, one positive example is *Bird sings before dog barks* and one negative example would be *Bird sings after dog barks*.

We provide the model with two positive text examples and two negative text examples for each text description containing the previously defined *future* and *past* temporal textual cues. Positive and negative text examples can be generated once, before training the model. We searched for the temporal cues we are interested in and automatically generated multiple positives and negatives by changing the temporal cues and/or the ordering of the sounds.

The contrastive loss for each query and a margin $\alpha$ is:

$$\mathcal{L}_{tt} = \frac{1}{2N} \sum_{n=1}^{N} \sum_{k=1}^{2} \max(0, \alpha - \text{pos\_sim}_{nk} + \text{neg\_sim}_{nk}), \quad (3)$$

where $\text{pos\_sim}_{nk}$ is the similarity between the $n$-th query and its $k$-th positive example, $\text{neg\_sim}_{nk}$ is the similarity between the $n$-th query and its $k$-th negative example. Our full loss then becomes:

$$\mathcal{L} = \mathcal{L}_{ta} + \lambda\mathcal{L}_{tt}. \quad (4)$$

In our experiments that use the text-text loss, we set $\lambda = 10$, $\alpha = 0.2$.

We now evaluate the same model pre-trained on WavCaps and finetuned on SynCaps but employing our additional loss. In Tab. 6, we observe that the model performs better on the original test set compared to Tab. 5, whilst at the same time showing a big drop in performance on the 'reversed' and 'replaced' data. This shows that employing a simple additional loss can help the model better understand time, at least in the synthetic controlled setting.

## 5 CONCLUSION

In this work, we dissected temporal understanding capabilities of current state-of-the-art text-audio model. We first analysed how well-suited AudioCaps is as a training and evaluation dataset for the temporal understanding of events. We then proposed a new synthetic dataset, concluding that indeed models fail to use the temporal cues even when the data is clean. Lastly, we propose a simple loss that results in better text-to-audio retrieval results on SynCaps, whilst also putting more emphasis on the temporal content of the audio and text data.

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
