# OpenReview forum: "Dissecting Temporal Understanding in Text-to-Audio Retrieval"
_acmmm.org/ACMMM/2024/Conference — MM2024 Poster_

### Official Review · Reviewer_CPdu · 2024-05-11

**Rating:** 3
**Confidence:** 3

**Summary:**

- The paper explores the temporal understanding of text-to-audio retrieval models
- The authors analyze this using the Audiocaps dataset and show that it is not well suited for this task
- Hence they introduce a more refined and uniform version of Audiocaps using LLM
- They further introduce synthetic data generated using the ESC-50 dataset.
- Finally, they introduce an additional temporal-relevant loss function

**Strengths:**

- Comprehensive analysis of the temporal understanding using AudioCaps is done.
- Simple but interesting way of analyzing the models' understanding of temporal by reversing and replacing words.
- A new dataset is introduced using the combinations from ESC-50 data
- The new loss function helps improve text-to-audio models' temporal understanding.

**Limitations:**

Major limitations:
- Lack of Novel Contributions:
  - The temporal understanding of text-audio models has already been addressed in previous research [1]. Therefore, the paper’s
     exploration of temporal understanding offers only slight novelty.
  - Although the paper conducts several comprehensive studies by swapping/replacing words, the contributions are relatively straightforward and simple. For example, manually selecting temporal words and reversing/swapping words to assess the model's temporal understanding.

- Non-Generalizable Findings:
  - The experiments are solely tested on the AudioCaps dataset, while there are several other relevant text-audio datasets, such as Clotho and WavCaps.
  - The specific words manually selected may only be representative of AudioCaps, but not necessarily applicable to text-audio datasets in general.

Minor limitations:
- The paper is hard to follow and the contributions of the work are not clearly stated. A revision with concise headings and structured writing could improve the paper's clarity.


[1] Wu et. al. AUDIO-TEXT MODELS DO NOT YET LEVERAGE NATURAL LANGUAGE

**Suitability:**

3

---

### Official Review · Reviewer_8JLR · 2024-05-23

**Rating:** 5
**Confidence:** 3

**Summary:**

The paper studies the ability of audio-language retrieval models to capture temporal relationships between different audio events occurring in a single recording. Authors evaluate pretrained encoders on two datasets, AudioCaps which naturally contains multiple audio events and textual descriptions of ordering, and ESC, where they synthetically curate differently ordered audio events. Authors also propose a new loss that can enforce models to capture temporal ordering of events.

**Strengths:**

1. The paper is clearly written and easy to understand, with the flow of experiments in order.
2. Although no novel models are introduced, experimentation with the given hypothesis is clear.
3. The paper seems to be technically correct.

**Limitations:**

1. Proposed method shown only for synthetic dataset, ideally should be repeated/evaluated on AudioCaps atleast for A-T retrieval
2. Event Localization metrics can be used to better exemplify the capability of audio encoders to capture temporal ordering

**Suitability:**

2

---

### Official Review · Reviewer_hGZH · 2024-05-24

**Rating:** 3
**Confidence:** 3

**Summary:**

The paper explores the ability of text-to-audio and audio-to-text models to understand and utilize temporal relationships in audio data. The paper addresses problems of existing approaches to understand and use temporal information. An commonly used dataset, AudioCaps is analyzed and improved in terms of temporal information. An improved model in presented and evaluated against the existing and improved datasets.

**Strengths:**

# Overall
* The paper is structured
* The paper fits to the conference

# Contributions
* The paper analyzes insufficiencies in the AudioCaps dataset and tries novel ways to improve the performance by an improved dataset  and a slightly changed model.
* AudioCaps dataset improved by better temporal captions
* Novel SynCaps dataset, synthetic dataset
* The results support the thesis, that models fail to use temporal information
* A better Loss function

**Limitations:**

## Content-related weaknesses
* Line 47: The paper is solely based on a single source
* Line 68: The authors write that they change the audio encoder, which is not clear in the rest of the paper (only mention in Line 325-327) how it impacts the performance.
* Line 291-293: The use of GPT-4 to evaluate the correctness is questionable. To my best knowledge, GPT-4 performs bad at understanding temporal contexts.
* Overall, the criticisms of AudioCaps are understandable but weak in their reasoning
* The Loss function is only evaluated against SynCaps, but not evaluated against the AudioCaps (Future work)

## Formal issues and errors

* Line 88: _<Original description of Sound_1> before <Original description of Sound_1>_; I assume _<Original description of Sound_1> before <Original description of Sound_2>_ would be correct? Otherwise I do not understand.
* Line 184 and 509 have overflow
* Line 755, cite 20 missing data ana last visited

**Suitability:**

3

---

### Official Review · Reviewer_mGVn · 2024-05-28

**Rating:** 3
**Confidence:** 4

**Summary:**

The paper focuses on the issue of temporal understanding in text-to-audio retrieval models, especially the temporal ordering of sound sources. The authors utilize LLMs to generate synthetic novel text-to-audio datasets in controlled settings and also evaluate temporal ordering and semantic correctness in existing datasets.

**Strengths:**

* Usage of LLM as an oracle to evaluate the captions in AudioCaps in terms of correctness and completeness.
* Introduction of a synthetic dataset under controlled settings to evaluate the temporal ordering capabilities of text-to-audio retrieval models.
* Enforcing temporal ordering of events for existing models through text-based contrastive loss.

**Limitations:**

* Regarding the usage of LLM, have the authors considered a simple paraphrasing-based approach where a caption containing “followed by” is rewritten using the phrase “preceded by”? Example: “a dog barking followed by bird singing” changed to “bird singing preceded by a dog barking.”
* The proposed loss is shown only on the controlled synthetic data generated from ESC50. In terms of a general approach, the role of the augmented loss (text-to-text contrastive) should be evaluated in the case of **AudioCaps**.
    * Table 4 should include results with the augmented loss (text-audio + text-text) under reversed and replaced settings of the test dataset.
* It is not clearly mentioned how the examples labeled as **“wrong”** by LLM are utilized in the training process. Is there an iterative process involving correction by LLM to improve the generation of descriptive captions?
* No mention of hallucination by LLM while generating the descriptive captions for **AudioCaps** or ESC. Further,
* For **SynCaps**, the results are shown on a held-out validation set. Ideally, in the absence of a test set, the results should be reported using a k-fold cross-validation scheme.

**Suitability:**

3

---

### Meta-Review · Area_Chair_7ijU · 2024-07-04

**Recommendation:** Accept (Poster)
**Confidence:** 4

**Metareview:**

The paper tackles the temporal ordering of sounds, in the context of text-to-audio retrieval. Additionally, it introduces a synthetic text-audio dataset that provides a controlled setting for evaluating the temporal understanding of recent models.

Overall, the novelty of the proposed approach is limited but the experimentation with the given hypothesis is clear. As a result, the paper introduces a more refined and uniform version of the Audiocaps dataset using LLM. The paper fits the MM2024 requirement by using audio-text to some extent and performances are specifically designed on the AudioCaps. After considering the paper, the reviewer's comments, and the rebuttal I recommend 'accept (poster)' for the paper.

The reviewers highlight the following strengths and limitations:

Strengths:
1. Usage of LLM as an oracle to evaluate the captions in AudioCaps in terms of correctness and completeness.
2. Introduction of a synthetic dataset under controlled settings to evaluate the temporal ordering capabilities of text-to-audio retrieval models.
3. AudioCaps dataset improved by better temporal captions

Limitations:
1. The temporal understanding of text-audio models has been addressed in previous research to some extent
2. Further evaluations of the loss function would be better.